# SpatialEdit: Unlocking the spatial capability in Multimodal Large Language Model Driven Image Editing

## Abstract

Current instruction-guided image editing methods generally believes that incorporating powerful Multimodal Large Language Model (MLLM) can significantly enhance the understanding of complex instructions, thereby improving editing outcomes and generalization. However, even using an powerful MLLM model such as GPT4V, disappointing results are observed when instructions involve simple spatial information such as "change the color of clothes of the leftmost person to red". Our theoretical analysis suggests that both the training strategy and the model aggregation manner in the current paradigm may contribute to unsatisfactory spatial image editing capabilities. Consequently, we propose the SpatialEdit framework, featuring a two-stage training approach and a novel data engine where questions and instructions are enriched with spatial information. Further theoretical analysis of our method reveals its ability to increase proficiency in both spatial editing and general image editing tasks. We create a benchmark to evaluate spatial editing ability. We conduct zero-shot image editing experiments on various datasets and our method achieves SOTA results on several key metrics.

## 1 Introduction

Image editing plays a pivotal role in various multimedia applications, enabling users to customize images to meet their specific preferences and needs. From basic color adjustments Gatys et al. (2016); Zhang et al. (2016) to intricate semantic alterations Zhu et al. (2016), image editing techniques are widely utilized in various fields. Instruction-guided image editing has emerged as a popular research area Li et al. (2020a); El-Nouby et al. (2019a); Fu et al. (2020a), offering a practical and intuitive approach where direct human commands dictate specific aspects of image manipulation.

The recent substantial work in the image editing community demonstrates confidence in the ability of MLLM to handle image editing tasks with complex instruction Zhang et al. (2023a); Koh et al. (2023); Liu et al. (2023b); Zhu et al. (2023). Recent mainstream efforts Fu et al. (2023); Ge et al. (2024) are also often focused on integrating MLLM into image editing tasks.

However, we find that the performance of existing MLLM-driven image editing models is not as good as imagined, especially in following spatial instructions, i.e., instructions with spatial information. For example, `remove apples on the kid's left hand side`. As illustrated in Figure 1, even advanced models such as DALLE3 powered by GPT4V 202 (2023) encounter difficulties in following spatial instructions.

In real-world scenarios, instructions often include spatial information, especially when dealing with complex scenes containing multiple subjects and intricate details Yang et al. (2024b). For instance, when editing a family photograph, instructions involving spatial information are often more direct and precise, such as referencing "the second individual from the left in the first row," rather than relying solely on descriptive attributes of the subject itself. Thus, We need to rethink the reasons why MLLM-driven methods performs poorly in spatial editing tasks.

To solve the problem, we conduct theoretical analysis of the current MLLM-driven image editing paradigms, and identify two possible factors contributing to the unsatisfactory performance of MLLM-driven methods. **Firstly**, contrary to intuition, incorporating MLLM to enhance the perfor-

| Input | MGIE | GPT4V | SEED-X | SpatialEdit |
|-------|------|-------|--------|-------------|

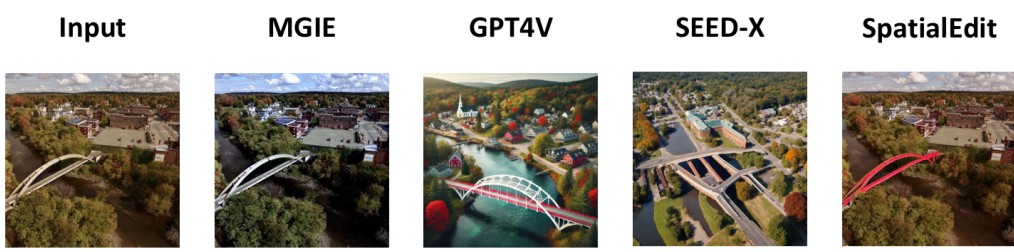

**Instruction**: Turn the **connecting area between the two land spaces** into red.

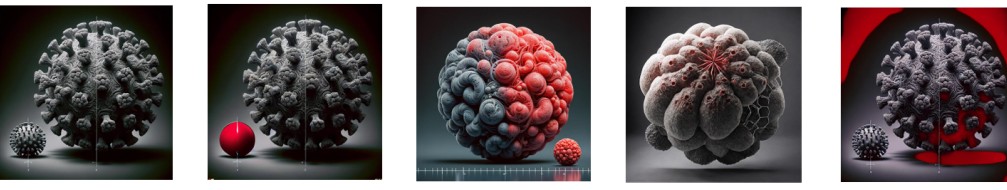

**Instruction**: Change the background of the **bigger ball** into red.

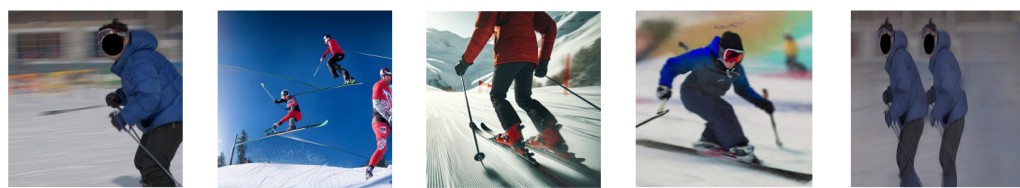

**Instruction**: Place this man in motion **at the next moment** he will arrive at.

Figure 1: Results of different methods in spatial editing tasks. Given instructions contain spatial information, even GPT4V struggle to follow spatial instructions. However, our 7B model outperforming the 17B Seed-X and GPT4V. More visualization results are shown in Figure 4 in the Appendix.

mance of spatial editing tasks is not a free lunch. Our analysis in §4.1 reveals potential over-fitting risks associated with integrating the MLLM with downstream edit heads and adapters, which may compromise model generalizability and lead to unsatisfactory editing outcomes. In Table 4, we find that existing datasets lack samples with high-quality spatial information. When the training samples with high-quality spatial instructions is limited, it further exacerbates the over-fitting risk in the spatial editing task. **Secondly**, as analyzed in §4.2, current paradigm fails to alter attention pattern flexibly. This limitation stems from the fact that only embeddings are trainable in current MLLM-driven methods.

To unlock the editing capacity in following spatial instructions, we introduce the SpatialEdit framework. This framework encompasses a data engine for high-quality data generation with spatial information, along with a novel two-stage training method. Specifically, to avoid the risk of overfitting, we created a data engine which can generate high-quality visual question answering (VQA) data and spatial image editing data at scale. Furthermore, to address the limitation of inflexible attention-pattern adjustment, we unfreeze the attention layer of MLLM during training, termed as attention tuning. In the initial training phase, we employ attention-tuning and use VQA data generated by our data engine to bolster the spatial understanding capabilities of MLLMs. Subsequently, the second training phase, we leverages the spatial image editing data generated by our data engine to enhance the capabilities of spatial editing.

In addition,we provide theoretical proof in §6 demonstrating that models trained on spatial data generated by our data engine consistently outperform those without such training, both in general editing tasks and in spatial editing tasks.

Our main contributions are summarized as follows:

- We provide theoretical insights on the underlying reasons of the unsatisfactory performance of existing MLLM-driven image editing methods.
- We propose a data engine which can automatically generate spatial VQA data and spatial image editing data given an image.
- We introduce a novel two-stage training approach. In the initial stage, our attention-tuning method enhances the spatial understanding capabilities of the MLLM, while the subsequent stage focuses on refining both spatial comprehension and editing proficiency.
- The efficacy of our SpatialEdit framework is rigorously validated through both theoretical analysis and experimental validation, which establish a significant advancement in MLLM-driven image editing.

## 2 RELATED WORK

Instruction-guided image editing has attracted considerable attention, providing a specifically intuitive way to edit an image through textual prompts provided by the user. Earlier methods using GAN frameworks Reed et al. (2016) encountered challenges in achieving broad applicability and realism Li et al. (2020b); El-Nouby et al. (2019b); Fu et al. (2020b). Diffusion models Ho et al. (2020); Saharia et al. (2022); Rombach et al. (2022) emerged as promising alternatives, offering more flexible image transformations by controlling cross-modal attention maps based on global captions Kawar et al. (2023); Hertz et al. (2023); Meng et al. (2022). While these studies are typically constrained to global editing capabilities Bar-Tal et al. (2022a); Crowson et al. (2022), local image editing methods have also been explored, enabling precise modifications while preserving image integrity Nichol et al. (2022); Avrahami et al. (2022); Bar-Tal et al. (2022b); Couairon et al. (2023).

MLLM-driven image editing methods incorporate MLLMs to address the limitation of traditional methods. However, the frozen CLIP text encoders, while proficient in processing static descriptions, may lack the capacity to facilitate crucial transformations in editing tasks, potentially leading to or imprecision in instruction comprehension. Recent approaches have leveraged MLLMs Fu et al. (2023); Ge et al. (2024) to address this challenge and have demonstrated remarkable capabilities in natural language understanding and generation tasks. Current MLLM-driven methodology still face a critical limitation in following spatial instructions. We aims to solve the problem by introducing SpatialEdit.

## 3 PRELIMINARY

In this section, we provide some necessary preliminaries and notations to aid understanding. Due to page limitations, certain other concepts (the self-attention architecture) can be referred to in E.

### 3.1 COMPONENT-STACKING PARADIGM

Current MLLM-driven image editing methods Fu et al. (2024) operate under the component-stacking paradigm, where multiple components are stacked to form the pipeline structure, as illustrated in the model training part in Figure 3. Mathematically, this paradigm can be represented as follows:

$$\mathbf{X}_{img}^{trainable} = \text{Ada}_{img}(\text{Enc}(\mathbf{M})), \tag{1}$$

$$[\mathbf{E}_{img}, \mathbf{E}_{txt}] = \text{H}_{lm}(\text{MLLM}(\mathbf{X}_{task}^{fix}, \mathbf{X}_{prompt}^{trainable}, \mathbf{X}_{img}^{trainable}, \mathbf{X}_{txt}^{fix})), \tag{2}$$

$$\mathbf{M}_o = \text{Diff}_{edit}(\text{H}_{edit}(\mathbf{E}_{img}), \mathbf{E}_{txt}, \mathbf{M}), \tag{3}$$

where $\mathbf{M}$ is the raw image, $\text{Enc}(\mathbf{M})$ is a sequence of visual features produced by a fixed pre-trained image encoder such as CLIP Radford et al. (2021), while $\text{Ada}_{img}$ denotes a trainable image adapter responsible for adapting the dimension of visual features generated by the pre-trained image encoder to match the input dimension of MLLM Radford et al. (2021). $\mathbf{X}_{task}^{fix}$ is the embedding sequence of a fixed prompt indicating the task. For image editing task, the prompt is "What will it be like if." $\mathbf{X}_{prompt}^{trainable}$ is the embedding sequence of soft prompt. Soft prompts are not human readable. They are learnable embeddings that can be optimized for a task. $\mathbf{X}_{img}^{trainable}$ is a sequence of visual features. $\mathbf{X}_{txt}^{fix}$ is the embedding sequence of the editing instruction such as "change the clothes color

of the leftmost person into read." $H_{lm}$ and $H_{edit}$ correspond to the language modeling and edit head, respectively. $H_{lm}$ refines the language output generated by MLLM and projects the output into both language and visual modalities $\mathbf{E}_{img}$ and $\mathbf{E}_{txt}$, with $\mathbf{E}_{img}$ further projected into a sequence of image condition embeddings by $H_{edit}$. $\mathbf{E}_{txt}$ is a sequence of language condition embeddings. Both $\mathbf{E}_{txt}$ and $H_{edit}(\mathbf{E}_{img})$ provide guidance for an conditional diffusion model $\text{Diff}_{editor}$. Consequently, the output of the conditional diffusion model, denoted as $\mathbf{M}_o$, represents the edited image.

According to Eq 1, 2, and 3, there are many stacked component pairs. For example, the edit head is stacked on the MLLM, and the diffusion model is stacked on the edit head.

Notably, during training, the MLLM remains frozen, while only the adapters and heads, namely $\text{Ada}_{img}, H_{edit}, H_{lm}$, and $\text{Diff}_{editor}$, are trainable.

## 4 THEORETICAL ANALYSIS

As shown in Figure 1, even utilizing a powerful MLLM such as GPT4V 202 (2023) and a diffusion model such as Dalle3 as core components, current MLLM-driven methods demonstrate a notable deficiency in following spatial instructions, leading to unsatisfactory editing outcomes. This observation underscores the need for thorough theoretical analysis.

Firstly, in §4.1, we reveals that employing MLLM into an image editing model might not be a free lunch because the component-stacking mechanism (e.g., the language modeling head is stacked on the MLLM, and the diffusion model is stacked on the MLLM,) exacerbates the risk of over-fitting, might diminishing the overall model's performance in image editing tasks. This contrasts with the widely held view that introducing MLLM would enhance the performance of any task requiring complex language comprehension capabilities.

In §4.2, we uncover that the current training strategy which only tunes prompt embeddings and image embeddings hinder the MLLM from learning spatial relationship, might limiting the MLLMs' capacity to guide the diffusion model towards desired editing outcomes.

### 4.1 RISK OF OVER-FITTING

In this section, we investigate the Rademacher complexity of the stacked component under component-stacking paradigm outlined in §3.1. The Rademacher complexity serves as a pivotal metric for assessing a model's maximum capability at fitting random noise during training. An increase in its value signifies a larger model capacity, while heightening the risk of over-fitting.

**Theorem 1** *Let $I = [n]$, $\mathcal{F}, \mathcal{G}$ be two countable function classes, $\mathcal{X}, \mathcal{T}$ be their domains, respectively, and $\mathbf{0} \in \mathcal{X}$. Suppose for any $f \in \mathcal{F}, g \in \mathcal{G}$, $f, g$ are Lipschitz and $\sup_{f \in \mathcal{F}} \|f\|_{Lip} \leq L_f < \infty$, $\sup_{g \in \mathcal{G}} \|g\|_{Lip} \leq L_g < \infty$, the rademacher complexity of the composited function class $\mathcal{F}(\mathcal{G})$ is*

$$b(\mathcal{F}(\mathcal{G})) \leq \lambda[L_f b^\tau(\mathcal{G}, \mathcal{T}) + L_g b^\tau(\mathcal{F}, \mathcal{X})], \tag{4}$$

*where $\lambda$ is a universal constant, $\|\cdot\|_{Lip}$ is the lipschitz constant, $b(\cdot)$ is the rademacher complexity, $b^\tau(\mathcal{F}, \mathcal{X})$ is the empirical rademacher complexity defined as:*

$$b^\tau(\mathcal{F}, \mathcal{X}) := \sup_{s,t \in \mathcal{X}} E\left[\sup_{f \in \mathcal{F}} \frac{\sum_{i \in I} \epsilon_i(f(s_i) - f(t_i))}{\|s - t\|}\right], \tag{5}$$

*where $\epsilon_i$ are a sequence of i.i.d rademacher variables.*

The Rademacher complexity Truong (2024) signifies a model's maximum capability to approximate to random noise, representing an inherent upper limit. Hence, it is important to analyze the upper bound of Rademacher complexity. Specifically, we analyze the Rademacher complexity of a composited function class $\mathcal{F}(\mathcal{G})$. In current component-stacking paradigm mentioned in §3.1, there are many instances of composited functions. For example, the MLLM $\in \mathcal{G}$, and the language modeling head $H_{lm} \in \mathcal{F}$; the edit head $H_{edit} \in \mathcal{G}$, and the diffusion model $\text{Diff}_{editor} \in \mathcal{F}$. Eq 4 implies that the Rademacher complexity of a stacked component is bounded by weighted the empirical Rademacher complexities of individual constituent components. Specifically, the weights are the upper bound of the Lipschitz constants. Typically, the Lipschitz constant tends to be large

Gouk et al. (2021). Thus, the current component-stacking paradigm is likely to exhibit pronounced increase in Rademacher complexity, might increasing the risk of over-fitting, consequently, might yielding unsatisfactory image editing outcomes.

Currently, the academic community has many methods to address over-fitting. Mainstream methods suggest that increasing the quantity as well as the quality of data can enhance model performance and alleviate over-fitting Thompson et al. (2022). However, the creation of such scalable and high-quality datasets can be expensive.

### 4.2 ATTENTION SCORES UNDER EMBEDDING TUNING

MLLM plays an important role in MLLM-driven methods because it is expected to provided good image and language conditions for the conditional diffusion model. In current training strategy, only prompt embeddings $\mathbf{X}_{prompt}^{trainable}$ and $\mathbf{X}_{img}^{trainable}$ image embeddings are trainable, while the the entire MLLM is frozen. We term this training strategy as embedding tuning.

The attention score of $x_i$ attending to $x_j$ under embedding tuning is formulated as follows,

$$A_{ij}^{et} = \frac{\exp\left(\frac{\tau_t}{\sqrt{d_k}} x_i^T W_Q^T W_K x_j\right)}{\sum\limits_{z=0}^{n_t} \exp\left(\frac{\tau_t}{\sqrt{d_k}} x_i^T W_Q^T W_K x_z\right) + \sum\limits_{r=n_t}^{p} \exp\left(\frac{\tau_t}{\sqrt{d_k}} x_i^T W_Q^T W_K x_r\right)} \tag{6}$$

$$= A_{ij} \sum\limits_{r=n_t}^{p} A_{ir}^{et}. \tag{7}$$

where $A_{ij}$ defined in Eq 11 in the Appendix E, denoting the attention score of $x_i$ attending to $x_j$ before the embedding tuning, $A_{ij}^{et}$ represents the score after embedding tuning, $n_t$ is the number of trainable embedding tokens, $p$ is the number of tokens in the input. $d_{in}$ denote the dimensionality of token embeddings, and $d_k$ is the dimensionality of key value. $\tau_t$ is the inverse temperature parameter with $\tau_t > 0$, $x_i \in \mathbb{R}^{d_{in}}$ is a token embedding.

As shown in Eq 14 in the Appendix E.1, the attention score after embedding tuning can be written as a form of the attention score before embedding tuning ($A_{ij}$) multiple by a weighted scalar. The attention score of $x_i$ attending to different $x_j$ is scaled equally, indicating the rank of the importance of $x_j$ to $x_i$ is unaltered. For example, before embedding tuning, the most important token to $x_i$ is $x_k$, then after embedding tuning, the most important token to $x_i$ is still $x_k$. However, an effective training process should allow for varying attention scores to effectively encode the image and instruction. For example, if the editing instruction is "change the hair of the leftmost person to red", the attention scores of word token "leftmost" attending to visual tokens which are related to the leftmost person in the image should be higher than others. But under the embedding tuning, altering attention scores is less flexible.

The result can be easily extend to multi-head situation, because the problem affect each head equally. The result can be easily extend to the network with multiple self-attention layers. The detailed analysis of both cases can be found in the Appendix E.2.

The outputs of MLLMs are expected to provide good image and language guidance to the diffusion model. The ineffective learning of attention layers might leads to bad image and language guidance.

## 5 METHOD

To address the issues revealed by the analysis, we propose SpatialEdit which consists of a data engine and a novel training method. Specifically, to mitigate the risk of over-fitting in the component-stacking paradigm as revealed in §4.1, we design a data engine can automatically generate spatial VQA data and spatial image editing data given a standard 2D image. This engine is designed to produce a substantial volume of data, limited only by the availability of input images. These QA pairs and image editing pairs are enriched with spatial information to facilitate comprehensive training of the entire pipeline, including the MLLM and diffusion editing head.

On the other hand, to address the issue raised in §4.2 where the existing training strategy struggles to enable an MLLM to generate correct and explicit conditions with spatial information, we develop

a training technique called Attention tuning and use it in our two-stage training method. The first stage of training can flexibly alter the attention scores between different tokens. The second stage training aims to improve the image editing capabilities in following spatial instructions.

In this section, we introduce our data engine in §5.1 and the two-stage training method in §5.2. To the best of our knowledge, currently there is no suitable benchmark that focuses on evaluating the image editing capabilities in following spatial instructions, hence we propose a new benchmark called SpatialEval in §5.3.

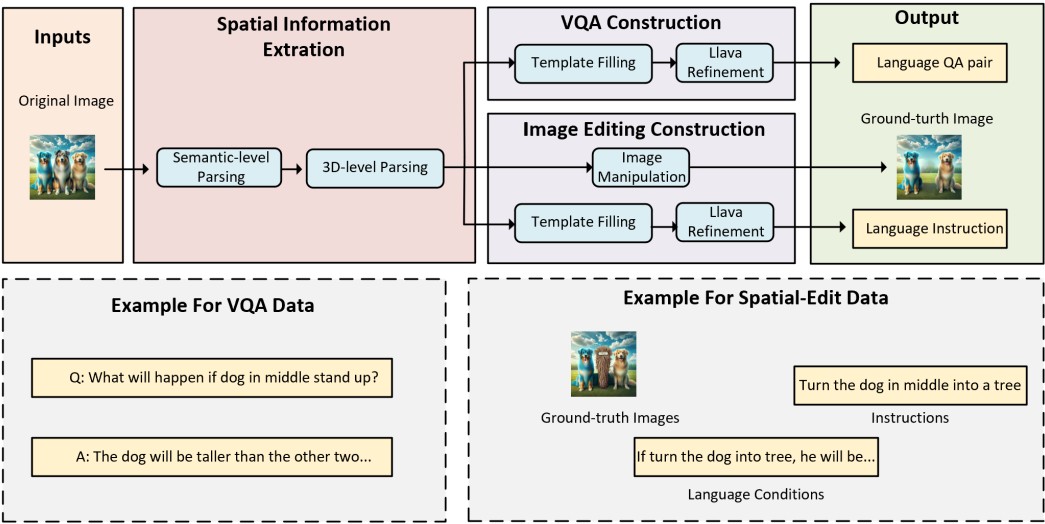

Figure 2: Pipeline of our data engine.

## 5.1 DATA ENGINE

Our theoretical analysis §4.1 reveals that the key to enabling MLLM-driven models to exhibit superior performance lies in the additional introduction of high-quality training data. However, from the statistical data in our Table 4, we found that the data generated by the existing data pipeline lacks spatial information, suggesting the need for a new data engine to construct image editing data that incorporates spatial information.

Moreover, since the MLLM is the core component of the entire pipeline responsible for understanding spatial relationships and controlling image editing, it is reasonable to construct data to train the MLLM before the end-to-end training on spatial image editing data. This allows the model to gain stronger spatial understanding, which can help our model learn better in spatial image editing tasks.

Therefore, our data engine should not only construct end-to-end image editing data but also generate VQA data for MLLM training. We first introduce raw spatial information extraction in §5.1.1, and then the construction of the VQA data in §5.1.2 and the construction of spatial image editing data in §5.1.3.

### 5.1.1 SPATIAL INFORMATION EXTRACTION

To enrich the questions (used to construct VQA data) and instructions (used to construct spatial image editing data) with spatial information, we first extract raw spatial information from the images. The whole pipeline of our data engine is shown in Figure 2. Initially, we conduct semantic-level parsing, where images from the source dataset Brooks et al. (2023a) are fed into the LLava-1.6 Liu et al. (2023a) which is responsible for recognizing and describing the content within the images and predicting future motion trends, forming a triplet [object, description, trend]. For example, the black dog in the original image of Figure 2 can be represented as Object:"dog", Description:"with black fur, sitting", Trend: "sticking its head out".

Then, we conduct a 3D-level parsing based on the results obtained above. The triplet is sent to GroundSAM Ren et al. (2024), which combines groundingDino Liu et al. (2023d) and Segment Anything Kirillov et al. (2023) for generating image segmentation masks with label of its category. Then the Depth anything Yang et al. (2024a) model is used for depth estimation and point cloud construction. The point clouds are segmented using the RANSAC Fischler and Bolles (1981) algorithm, forming a quintuple [Object, Description, Trend, Point cloud, Mask] where the point clouds are represented by a set of 3D coordinates, and the mask is represented as a 2D array. The quintuple is further used to calculate more spatial information including position, contour size, volume, and relative distance, ultimately forming a list [Object, Description, Trend, Position, Contour, Volume, Distance]. After conducting above extraction steps, we extract the raw spatial information of an image.

### 5.1.2   VQA DATA CONSTRUCTION

The VQA data generated by our data engine is used to support the first stage training of enhancing the spatial understanding capability of the MLLM. The construction of the VQA data includes two operations: filling in the template with raw spatial information and refining questions with LLava-1.6 Liu et al. (2023c). The template designed for VQA data effectively transforms raw spatial information extracted in §5.1.1 into question-answer pairs. An detailed example is provided in Appendix B.1.

There are various types of templates which describes depth, relative and absolute positions, contour size, actual volume, and motion trends, which comprehensively cover the scope of spatial information. More details about templates are shown in Appendix K. The tentative QA pairs are then fed into Llava Liu et al. (2023c) again for diversified expression.

### 5.1.3   SPATIAL IMAGE EDITING DATA CONSTRUCTION

Spatial image editing data is designed to train models with better editing capabilities in following spatial instructions and to generate more explicit language conditions. Each sample in the data represented as a tuple [instruction, language condition, ground-truth image].

Firstly, we generate the instructions. The generation of instruction includes two parts, filling in the template and refining with llava-1.6. The idea of filling template is mostly the same as we illustrated in §5.1.2, and a detailed example is shown in appendix B.2 Some example templates are available in the Appendix K.

Subsequently, we construct language conditions. Note that the language condition are different from the language condition embeddings $\mathbf{E}_{txt}$ in §3.1. Language condition is textual. However, a sequence of language condition embeddings can be decoded into text using a tokenizer. Hence language conditions serve as gold labels for the MLLM to generate better language condition embeddings. We use the prompt we designed for Llava-1.6 Liu et al. (2023a) to ask Llava-1.6 to generate detailed description of the expected editing result corresponding to the instruction and current image. For example, a proper language condition can be "the dog with black fur, which on the left of golden fur dog, is replaced with a brown tree trunk".

Finally, we generate the ground-truth images in different manners for five types of instructions, including removing a object, changing the size of an object, visualizing the moving object at next moment, replacing one object with another, and change the texture or color of an object. For the removing instruction aimed at removing a specific object, we employ the object's mask as a guide for the stable diffusion Podell et al. (2023) to the generate of image where the targeted object has been successfully removed. Due to space limitation, the details of rest instructions can be found in Appendix M.

### 5.2   TRAINING STRATEGY

### 5.2.1   STAGE I

The first stage of training aims to enhance the spatial understanding capabilities of MLLM. We train the MLLM on the VQA data generated by our data engine. The analysis in §4.2 reveals the shortcomings of embedding tuning, so we design a new training method called attention tuning

which allows attention layers in the MLLM to be trainable while keeps other layers in MLLM frozen to alter the weight of attention layer. Noted that the image adapter is still trainable to enable better representation of image inputs. The detailed training pipeline as well as model architecture are shown in Figure 3 in the Appendix.

### 5.2.2 STAGE II

The second stage of training aims to enhance the editing capabilities in following spatial instructions and capabilities in generating explicit and precise conditions. The second stage training is conducted on the image editing data generated by our pipeline. The objective function is defined as,

$$\mathcal{L} = \mathcal{L}_{condition} + \mathcal{L}_{edit}, \tag{8}$$

where the $\mathcal{L}_{condition}$ is cross-entropy loss between the output of language modeling head $H_{lm}$ of MLLM and language condition embedding tokenized by tokenizer in training data. $\mathcal{L}_{edit}$ is the classifier-free guidance diffusion loss between output of diffusion editor and ground-truth image. As shown in Figure 3, the trainable components are image adapter $Ada_{img}$, parameter of attention layers, language modeling head $H_{lm}$, edit head $H_{edit}$, and diffusion model $Diff_{edit}$.

### 5.3 SPATIALEVAL BENCHMARK

To the best of our knowledge, currently there is no suitable benchmark that focuses on evaluating the image editing capabilities in following spatial instructions. Hence, it is necessary to develop a new benchmark. Therefore, we propose the SpatialEval benchmark, a benchmark designed to assess the model's image editing capabilities in following spatial instructions.

The benchmark consists of 90 images, among which 30 are from OpenImages Kuznetsova et al. (2020), 30 are from COCO2017 Lin et al. (2015), and 30 are from DAQUAR Malinowski and Fritz (2014). Since we only evaluate the model's zero-shot capabilities, the benchmark serve as the test set, and there is no training set, namely the Ground-truth edited images are not provided. Each image have three editing instructions written by a human annotator. These instructions mainly are related to spatial perception, spatial reasoning, and spatial visualization capabilities. We propose three evaluation metrics for the benchmark, namely, GPT-4V, GPT3.5, and human evaluation. Due to space limitation, the details of evaluation process can be found in the Appendix N.2.

## 6 THEORETICAL ANALYSIS

During inference, there exits some instructions which do not contain spatial information. We term this situation as the general editing task. If the instructions contain spatial information, we term this situation as the spatial editing task. In this section, we conduct a theoretical analysis of the performance of our method under general and spatial editing tasks.

The data generated by our data engine will provide language guidance with spatial information. If the model is trained on these data, language condition embeddings and image condition embeddings generated by the MLLM contain spatial information. In this case, we state that the model is trained on spatial conditions. If a model is not trained on such data, we state that a model trained on non-spatial conditions.

**Theorem 2** *For any spatial condition $c \in C$, parameter $\theta \in \Theta$, suppose it is easier to approximate distribution $q(x|c)$ than distribution $q(x)$ by only adjusting the embedding $E$ of $p_{\theta,E}(x)$, i.e. $\min_E D(q(x|c)\|p_{\theta,E}(x)) < \min_E D(q(x)\|p_{\theta,E}(x))$, where $D$ is a convex divergence. For the general editing task, the divergence between the gold distribution $q(x)$ and the distribution produced by the model trained on spatial conditions $p_{\theta_s^*,\phi_s^*}(x)$ satisfies:*

$$D(q(x)\|p_{\theta_s^*,\phi_s^*}(x)) < D(q(x)\|p_{\theta_n^*,E_n^*}(x)), \tag{9}$$

*where $\Theta$ include all measurable functions, $\theta_s^*$ and $\phi_s^*$ are optimal parameters and embeddings trained on spatial conditions, $\theta_n^*, E_n^*$ are optimal parameters and embeddings trained on non-spatial conditions.*

*For the spatial editing task, the divergence between the gold distribution $q(x|c)$ and distribution produced by the model trained on spatial conditions $p_{\theta_s^*, \phi_s^*}(x|c)$ satisfies:*

$$D(q(x|c)\|p_{\theta_s^*, \phi_s^*}(x|c) < D(q(x|c)\|p_{\theta_n^*, E_n^*}(x|c))). \tag{10}$$

Due to space limitation, the proof of this theorem is in Appendix G. Eq 9 demonstrates that during inference, even when the instruction does not contain explicit spatial information (for example, "change the color of the dog who is sticking out its tongue to blue"), models trained on explicit spatial conditions still outperform those without such training. This finding explains why our method outperforms others Fu et al. (2024) in widely used benchmark such as MagicBrush Zhang et al. (2023b), EVR Tan et al. (2019), and Ma5K Shi et al. (2021a), despite finishing those editing tasks do not explicitly require spatial capabilities. Some other examples presented in Figure 4 also demonstrate the ability of our framework to perform precise editing.

Eq 10 demonstrates that during inference, when the instruction contain explicit spatial information, models trained with explicit spatial conditions outperform those without such training.

In summary, our theory elucidates why our training data and approach enhances the overall image editing proficiency of our model, no matter in the general editing task and spatial editing task.

## 7 EXPERIMENTS

### 7.1 EXPERIMENTAL SETUP

**Compared Methods.** We compared our method with the following baselines. **InsPix2Pix** Brooks et al. (2023b), the SOTA in non-MLLM based image editing methods, which utilizes the CLIP Hessel et al. (2021) as the text encoder and the StableDiffusion Podell et al. (2023) the diffusion model. **MGIE** Fu et al. (2023), the SOTA in MLLM-driven image editing methods, which leverages the Llava Liu et al. (2023c) to interpret expressive instructions and provide explicit guidance for image manipulation. **LGIE**, a LLM based image editing model employs LLaMA-7B Touvron et al. (2023) to process expressive instructions derived solely from textual inputs. **Dalle3**, which utilizes the GPT4V 202 (2023) as the MLLM and Dalle3 Betker et al. ([n. d.]) as diffusion model for image editing tasks. **Seed-X** Ge et al. (2024) is a 17B MLLM that can perform image editing.

**Ablation Study.** **SpatialEdit**$_{\text{Finetune}}$ is the model only pretrained on IPr2Pr and further finetuned on 25K data from IPr2Pr (which is the same amount of the data generated by our data engine) following the training method proposed in Fu et al. (2023). **SpatialEdit w/o S1** is model trained using our method without the first stage training, **SpatialEdit w/o S2** denotes the model is trained using our method without the second stage training. **SpatialEdit w/o DE** is trained on 25K IPr2Pr data using attention tuning instead of the data generated by our data engine. Detailed description is shown in Appendix D

**Metrics.** We utilize L1, DINO, SSIM, CVS, and LPIPS Zhang et al. (2018) for assessing image difference and employ CTS Hessel et al. (2021) to measure the similarity between gold captions and edited images. The metric used in SpatialEval benchmark is defined in §5.3.

**Datasets.** Seed-X Ge et al. (2024) is trained on MagicBrush Zhang et al. (2023b), IPr2Pr Brooks et al. (2023a) and Seed-data-edit Ge et al. (2024), all other methods are trained on IPr2Pr Brooks et al. (2023b). Additionally, Our Method is further trained using our training strategy. The image source of our data is only from IPr2Pr Brooks et al. (2023b). To evaluate the zero-shot image editing ability, we evaluate methods on EVR Tan et al. (2019), GIER Shi et al. (2020), MA5k Shi et al. (2021a), MagicBrush Zhang et al. (2023b), and SpatialEval benchmarks. Our VQA data consists of 62K question-answer pairs generated for 10K images, while our spatial image editing data contains 25K high-quality data tuples. The statistics of other datasets are shown in Appendix J.

**Implementation Details.** We show implementation details in Appendix L.

### 7.2 SPATIAL IMAGE EDITING

We evaluate the spatial editing ability of different methods on SpatilEval benchmark. As shown in Table 1, our SpatialEdit framework which leverages Llava7B as the backbone MLLM achieves the SOTA performance across various evaluation metrics.

| Method | GPT4V↑ | GPT3.5↑ | Human↑ |
|---|---|---|---|
| InsPix2Pix Brooks et al. (2023a) | 0.28 | 0.22 | 0.31 |
| MGIE Fu et al. (2024) | 0.47 | 0.34 | 0.38 |
| Dalle3 Betker et al. ([n. d.]) | 0.51 | 0.32 | 0.36 |
| Seed-X Ge et al. (2024) | 0.45 | 0.32 | 0.38 |
| SpatialEdit$_{Finetune}$ | 0.41 | 0.22 | 0.32 |
| SpatialEdit$_{w/o\ DE}$ | 0.44 | 0.24 | 0.40 |
| SpatialEdit$_{w/o\ S1}$ | 0.66 | 0.47 | 0.50 |
| SpatialEdit$_{w/o\ S2}$ | 0.62 | 0.29 | 0.38 |
| SpatialEdit | **0.73** | **0.53** | **0.57** |

Table 1: Results on SpatialEval benchmark. MGIE and SpatialEdit use Llava-7B as the MLLM while Dalle3 uses GPT4V. Note that Seed-X is a 17B model while SpatialEdit is a 7B model.

| Method | EVR | | | GIER | | | MA5k | | | MagicBrush | | | |
|---|---|---|---|---|---|---|---|---|---|---|---|---|---|
| | L1↓ | DINO↑ | CVS↑ | L1↓ | SSIM↑ | CVS↑ | L1↓ | SSIM↑ | LPIPS↓ | L1↓ | DINO↑ | CVS↑ | CTS↑ |
| InsPix2Pix*Brooks et al. (2023a) | 0.189 | 67.8 | 81.4 | 0.144 | 57.5 | 86.6 | 0.176 | 58.9 | 0.359 | 0.101 | 71.5 | 85.2 | 29.3 |
| LGIE* Fu et al. (2024) | 0.159 | 69.7 | 82.0 | 0.152 | 56.9 | 87.0 | 0.144 | 64.6 | 0.327 | 0.084 | 80.9 | 88.9 | 30.1 |
| MGIE* Fu et al. (2024) | 0.163 | 71.5 | 81.7 | **0.135** | 59.2 | 88.6 | 0.133 | 66.3 | 0.298 | 0.082 | 82.2 | 91.1 | **30.4** |
| SpatialEdit$_{Finetune}$ | 0.237 | 59.9 | 75.6 | 0.210 | 49.6 | 85.2 | 0.206 | 52.6 | 0.399 | 0.094 | 75.1 | 85.8 | 22.7 |
| SpatialEdit$_{w/o\ DE}$ | 0.222 | 69.4 | 78.1 | 0.203 | 50.2 | 85.5 | 0.186 | 55.7 | 0.352 | 0.092 | 80.1 | 87.8 | 24.6 |
| SpatialEdit$_{w/o\ S1}$ | 0.171 | 70.0 | 78.6 | 0.186 | 66.1 | 85.0 | 0.167 | **73.7** | 0.098 | 0.089 | 81.9 | 81.9 | 24.9 |
| SpatialEdit$_{w/o\ S2}$ | 0.270 | 57.2 | 69.7 | 0.209 | 52.8 | 81.3 | 0.158 | 71.1 | 0.160 | 0.098 | 67.2 | 82.9 | 25.9 |
| SpatialEdit | **0.153** | **79.8** | **83.0** | 0.164 | **68.0** | **90.8** | **0.131** | 73.6 | **0.096** | **0.057** | **90.0** | **93.5** | 28.0 |

Table 2: Zero-shot editing results. * denotes that the results are retrieved from Fu et al. (2024).

## 7.3 GENERAL IMAGE EDITING

We evaluate general image editing ability of different methods across various metrics on four datasets. These four datasets are designed for evaluating general image editing capabilities. As depicted in Table 2, SpatialEdit method mostly outperforms others across various metrics on all datasets in the zero-shot image editing task, indicating the general image editing proficiency of our model.

## 7.4 SPATIAL INSTRUCTION IN TRAINING DATA

We investigate the number of spatial instructions in the existing datasets and the data generated by our data engine. The results in Table 3 in Appendix show that, compared to other datasets where only a small number of instructions are classified as spatial instructions, our data has the highest proportion of spatial instructions.

## 7.5 DATA QUALITY

In order to further analyze the data engine, we conducted human evaluation from multiple perspectives to assess the quality of our data. We found that the quality of our automatically generated data is higher than most datasets, especially showing an advantage in spatial information. For detailed results, please refer to Appendix I.

## 8 CONCLUSION

Our theoretical analysis uncovers the potential causes behind the underperformance of the current MLLM-driven image editing paradigm in following spatial instructions. This insight serves as a catalyst for the development of SpatialEdit, a novel framework designed to address the above limitation. Through the integration of the data engine and a novel two stage training method, SpatialEdit significantly enhances the spatial capabilities of MLLMs. Theoretical analysis and experimental validation demonstrate that SpatialEdit not only achieves SOTA results in the zero-shot general editing task but also excels in spatial editing tasks.

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

## A  Detailed Illustration of Model Architecture

As illustrated in Figure 3, We construct SpatialEdit based on the model architecture of MGIE Fu et al. (2023), which includes an MLLM responsible for prompt understanding and guidance for the diffusion unit, as well as an edit head structure serving as an adapter and a diffusion editor based on diffusion.

## B  Examples on Data Construction

### B.1  VQA data

For example, given raw spatial information is [{ Object: dog, Description: black fur, Position: [0,0] },{ Object: dog, Description: golden fur, Position: [1,1] }], given one of the template in the template pools is [Q:"What position do $<item1><description1>$ from $<item2><description2>$" A: "$<item1><description>$ is at $<left/right>$ of $<item2><description>$"], filling in the template with above raw information leads to a tentative QA pair: [Q:"What position do dog with black fur to dog with golden fur? " A:"Right"].

Figure 3: Training of our spatialEdit framework. $N$ is the number of basic Transformer layers which mainly consist of an attention and a feed forward neural network in MLLM.

## B.2 SPATIAL IMAGE EDITING DATA

For example, given raw spatial information is [{ Object: dog, Description: black fur, Position: [0,0] }, { Object: dog, Description: golden fur, Position: [1,1] }], given a template for editing instruction is "change the $< item1 >< description >$ in $< position >$ of $< item2 >< description >$ into $< randomcolor >$", we want to construct an editing instruction, so we randomly choose two objects from spatial information i.e. "a black fur dog" and "a golden fur dog", then we fill in the template, obtaining the output "change the black fur dog in the left of golden fur dog into red". After that, we designed a prompt to ask llava-1.6 to refine the expression.

## C MORE CASES OF EDITED IMAGES

In Figure 4, we display results edited by our method as well as the results from other methods including IPr2Pr Brooks et al. (2023a), MGIEFu et al. (2024) and Seed-X Ge et al. (2024) (17B). When handling spatial instructions (and other complex instructions), both MGIE and IPr2Pr exhibit difficulties in accurately editing images as instructed. This often leads to either no discernible alterations or image distortions, and even produces edits that are totally different from the source images (e.g., significant disparity in image scenes and subjects compared to the source). In contrast, our SpatialEdit method excels in correctly adhering to instructions, consistently delivering the expected image edits with precision.

## D DETAILS ABOUT ABLATION STUDY

Our core contributions consist of three parts: the first part is the attention tuning technique, and the remaining parts are stage 1 and stage 2 where the model is trained on the data generated by the data engine. The comparison between **SpatialEdit**$_{\text{w/o DE}}$ and **SpatialEdit**$_{\text{Finetune}}$ demonstrates the effectiveness of the attention tuning technique itself, as it is conducted on an equal dataset and data volume. The effectiveness of stage 1 and stage 2, namely **SpatialEdit**$_{\text{w/o S2}}$ and **SpatialEdit**$_{\text{w/o S1}}$, will be shown by comparing them with **SpatialEdit**$_{\text{Finetune}}$ and the full version of SpatialEdit.

# E DETAILED ANALYSIS OF ATTENTION

In this section, we will provide a detailed explanation of the analysis and its generalization presented in §4.2 of our main body. Due to space limitations, we will present the complete content here.

A simplified decoder-only self-attention mechanism is Vaswani et al. (2023) applied across the entire sequence. Each attention block comprises $N_h$ heads. the attention score of each head $h$ is parameterized by query, key matrices: $W_Q \in \mathbb{R}^{d_k \times d_{in}}, W_K \in \mathbb{R}^{d_k \times d_{in}}$. Here, $d_{in}$ denote the dimension of input, and $d_k$ is the dimension of key value. For length of input $p$, the attention scores matrix $A \in \mathbb{R}^{p \times p}$ is defined as follows:

$$A_{ij} = \frac{\exp\left(\frac{\tau_t}{\sqrt{d_k}} x_i^T W_Q^T W_K x_j\right)}{\sum_{r=0}^{p} \exp\left(\frac{\tau_t}{\sqrt{d_k}} (W_Q x_i)^T (W_K x_r)\right)}, \tag{11}$$

where $\tau_t$ is the inverse temperature parameter, and $\tau_t > 0$, $x_i \in \mathbb{R}^{d_{in}}$.

## E.1 DETAIL ABOUT EMBEDDING-TUNING

In this section, we will provide a detailed derivation of the formulas mentioned in main body, as well as a more rigorous analysis.

According to preliminary, we have

$$A_{ij} = \frac{\exp\left(\frac{\tau_t}{\sqrt{d_k}} x_i^T W_Q^T W_K x_j\right)}{\sum_{r=0}^{p} \exp\left(\frac{\tau_t}{\sqrt{d_k}} x_i^T W_Q^T W_K x_r\right)}, \tag{12}$$

given that all token can be separated as trainable part and untrainable part, so we have:

$$A_{ij}^{et} = \frac{\exp\left(\frac{\tau_t}{\sqrt{d_k}} x_i^T W_Q^T W_K x_j\right)}{\sum_{z=0}^{n_t} \exp\left(\frac{\tau_t}{\sqrt{d_k}} x_i^T W_Q^T W_K x_z\right) + \sum_{r=n_t}^{p} \exp\left(\frac{\tau_t}{\sqrt{d_k}} x_i^T W_Q^T W_K x_r\right)}, \tag{13}$$

We then can write the attention as the following form:

$$A_{ij}^{et} = \frac{\exp\left(\frac{\tau_t}{\sqrt{d_k}} x_i^T W_Q^T W_K x_j\right) \sum_{r=n_t}^{p} \exp\left(\frac{\tau_t}{\sqrt{d_k}} x_i^T W_Q^T W_K x_r\right)}{A_{all}\left(\sum_{z=0}^{n_t} \exp\left(\frac{\tau_t}{\sqrt{d_k}} x_i^T W_Q^T W_K x_z\right) + \sum_{r=n_t}^{p} \exp\left(\frac{\tau_t}{\sqrt{d_k}} x_i^T W_Q^T W_K x_r\right)\right)} \tag{14}$$

$$= A_{ij} \sum_{r=n_t}^{p} A_{ir}^{et} \tag{15}$$

where $A_{all} = \sum_{r=0}^{p} \exp\left(\frac{\tau_t}{\sqrt{d_k}} x_i^T W_Q^T W_K x_r\right)$.

## E.2 DETAIL ABOUT MULTI-HEAD AND MULTI-LAYER

If each attention head in multi-head attention is plagued by the same issue, then the final linear combination of attentions cannot solve this problem either.

In analyzing transformer networks, multi-layer networks often present challenges for effective analysis. We can refer to previous work on the experimental results of prefix-tuning Petrov et al. (2024), which is a fine-tuning method that inserts some trainable tokens before the input of each layer. This undoubtedly provides flexibility similar to, or even greater than, what we can achieve in modifying the model. However, the analysis reveals that prefix-tuning in multi-layer transformer networks struggles to learn new tasks and change attention patterns effectively. We can generalize this conclusion, which is consistent with our experimental results.

# F    PROOF FOR THEOREM 1

We start the proof by replacing $\mathcal{F}(\mathcal{G})$ with $\mathcal{F}(T_{domain})$, where $T_{domain}$ is the range of $\mathcal{G}$, also the domain of $\mathcal{F}$.

Then, we divide the entire proof problem into studying the properties of the domain $\mathcal{X}$ and the properties of $\mathcal{F}(\mathcal{X})$. First, we study the $L_2$ diameter properties of $\mathcal{X}$. In the assumptions, we have constrained the Lipschitz property of functions in the family $\mathcal{G}$, that is, for any $f \in \mathcal{F}, g \in \mathcal{G}$, $f, g$ are Lipschitz and $\sup_{f \in \mathcal{F}} \|f\|_{\text{Lip}} \leq L_f < \infty$, $\sup_{g \in \mathcal{G}} \|g\|_{\text{Lip}} \leq L_g < \infty$.

Given that $f(x)$ is Lipschitz continuous, there exists a direct relationship between the $L_2$ diameter of $f(x)$ and its Lipschitz constant.

Firstly, the definition of Lipschitz continuity is given as follows: A function $f(x)$ is said to be Lipschitz continuous if there exists a constant $L$, such that for all $x_1, x_2$ within its domain, the following inequality holds:

$$\max_{x_1, x_2 \in D_m} \|f(x_1) - f(x_2)\|_2 \leq L \max_{x_1, x_2 \in D_m} \|x_1 - x_2\|_2, \tag{16}$$

where $L$ is the Lipschitz constant, and $\|x_1 - x_2\|_2$ denotes the $L_2$ (Euclidean) distance between $x_1$ and $x_2$.

Given a function $f(x)$, its $L_2$ diameter is defined as the maximum $L_2$ distance across the range of values that $f(x)$ takes within its domain. Specifically, considering a set of points $D$ within the domain, the $L_2$ diameter of $f(x)$ over $D_m$ can be expressed as:

$$\max_{x_1, x_2 \in D_m} \|f(x_1) - f(x_2)\|_2. \tag{17}$$

From the property of Lipschitz continuity, it is known that:

$$\Delta_2(D_m) \leq L \max_{x_1, x_2 \in D_m} \|x_1 - x_2\|_2, \tag{18}$$

where $L$ is the Lipschitz constant. This indicates that the $L_2$ diameter of $f(x)$ is constrained by its Lipschitz constant. In other words, the Lipschitz constant $L$ provides an upper bound that limits the maximum rate of change in the values of $f(x)$ within its domain. Consequently, the $L_2$ diameter of $f(x)$ cannot exceed the $L_2$ diameter of its domain $D$ multiplied by the Lipschitz constant $L$.

Then we start to analyse the rademacher complexity $b(\mathcal{F}(\mathcal{X}))$, we can introduce a corollary proven by previous workChu (2021):

**Collary 1 (Theorem 1.1 of Chu (2021))** *Suppose the conditions in Theorem 1.1 of Chu (2021) hold. Additionally, assume that the diameter $\Delta_2(\mathcal{X})$ is bounded by some constant $D$, i.e., $\Delta_2(\mathcal{X}) \leq D < \infty$. Then for any Lipschitz function $f$ in the countable function class $\mathcal{F}$ with Lipschitz constant $L$ bounded by some constant $L_f$, the complexity measure $b(\mathcal{F}(\mathcal{X}))$ is bounded as follows:*

$$b(\mathcal{F}(\mathcal{X})) \leq \lambda L_f b(\mathcal{X}) + \lambda D b^\tau(\mathcal{F}, \mathcal{X}), \tag{19}$$

*where $\lambda$ is a constant that incorporates the constants from the original bound in Theorem 1.1, $b^\tau(\mathcal{F}, \mathcal{X})$ is defined as*

$$b^\tau(\mathcal{F}, \mathcal{X}) := \sup_{s,t \in \mathcal{X}} E\left[\sup_{f \in \mathcal{F}} \frac{\sum_{i \in I} \epsilon_i (f(s_i) - f(t_i))}{\|s - t\|}\right]. \tag{20}$$

The Rademacher complexity measures the complexity of the hypothesis space, while the empirical Rademacher complexity is the instantiation of this measure on a specific dataset. Therefore, they can be conveniently transferred, i.e., we can substitute $b(\mathcal{X})$ with $b^\tau(\mathcal{G}, \mathcal{T})$. We can also scale the $D$ using the previously obtained in Eq. 18 where in the study of $g$, $L$ in Eq. 9 is $L_g$. Using the notation in theorem 1, we have:

$$b(\mathcal{F}(\mathcal{G})) \leq \lambda[L_f b^\tau(\mathcal{G}, \mathcal{T}) + L_g b^\tau(\mathcal{F}, \mathcal{X})], \tag{21}$$

Note that for the fixed input space, its $L_2$ diameter is always a constant, so we can merge the $L_2$ diameter of input space, namely, the $\max_{x_1, x_2 \in D_m} \|x_1 - x_2\|_2$ into $\lambda$.

We finish the proof.

| Dataset | Spatial Instruction | Non-Spatial Instruction |
|---------|--------------------|-----------------------|
| IPr2PrBrooks et al. (2023a) | 7.3% | 92.7% |
| MagicBrush Zhang et al. (2023b) | 3.3% | 96.7% |
| MA5K Shi et al. (2021b) | 8.3% | 91.7% |
| Seed-X Ge et al. (2024) | 6.7% | 93.3% |
| SpatialEdit | 44.3% | 55.7% |

Table 3: Assessment on the percentage of spatial instruction in some Image Editing datasets.

## G  PROOF FOR THEOREM 2

We can prove that in generation targets that require spatial conditions, our training includes the objective of generating explicit spatial conditions, thereby enabling better generation. Furthermore, in generation targets that do not require spatial condition information, i.e., traditional image editing tasks, we can also demonstrate that the training we conduct, which has clear spatial condition information constraints, can help the model generate better.

Theorem 2 have two cases, the case that target generation object is not restricted by spatial condition, namely $D(q(x)\|p_{\theta_s^*,\phi_s^*}(x)) < D(q(x)\|p_{\theta_n^*,E_n^*}(x))$, have been proven by previous work Bao et al. (2022), so we concentrate on the proof of the situation that target generation object is restricted by spatial conditions, namely $D(q(x|c)\|p_{\theta_s^*,\phi_s^*}(x|c)) < D(q(x|c)\|p_{\theta_s^*,E_s^*}(x))$.

According to the definition of $\theta_n^*$ and $E_n^*$, we have

$$\min_{\theta\in\Theta,E\in\phi} D(q(x)\|p_{\theta,E}(x)) = \min_{\theta\in\Theta,E\in\phi} D(\mathbb{E}_{q(c)}q(x|c)\|p_{\theta_n^*,\phi_n^*}(x)) \tag{22}$$

$$\leq \mathbb{E}_{q(c)} \min_{\theta\in\Theta,E\in\phi} D(q_{(x|c)}\|P_{\theta,E}(x)) \tag{23}$$

$$= \mathbb{E}_{q(c)} D(q(x|c)\|P_{\theta_n^*,E_n^*}(x)). \tag{24}$$

From Equation 22 to Equation 23, we used the Jensen's inequality. Then according to the definition of $\theta_s^*, \phi_s^*$, we have:

$$\mathbb{E}_{q(c)} D(q(x|c)\|p_{\theta_s^*,\phi_s^*}(x|c)) = \min_{\theta,\phi} \mathbb{E}_{q(c)} D(q(x|c)\|p_{\theta,\phi}(x|c)) \tag{25}$$

$$= \min_{\theta,\phi} \mathbb{E}_{q(c)} D(q(x|c)\|p_{\theta,\phi(c)}(x|c)) \tag{26}$$

$$= \min_{\theta} \mathbb{E}_{q(c)} \min_{\phi(c)} D(q(x|c)\|p_{\theta,\phi(c)}(x|c)) \tag{27}$$

$$= \min_{\theta} \mathbb{E}_{q(c)} \min_{E} D(q(x|c)\|p_{\theta,E}(x)). \tag{28}$$

According to assumption:

$$\min_{\theta} \mathbb{E}_{q(c)} \min_{E} D(q(x|c)\|p_{\theta,E}(x)) < \min_{\theta\in\Theta,E\in\phi} D(q(x)\|p_{\theta,E}(x)). \tag{29}$$

So combining above equations, we have:

$$\mathbb{E}_{q(c)} D(q(x|c)\|p_{\theta_s^*,\phi_s^*}(x|c)) < \mathbb{E}_{q(c)} D(q(x|c)\|P_{\theta_n^*,E_n^*}(x)). \tag{30}$$

We finished the proof.

## H  EVALUATION OF THE NUMBER OF SPATIAL INSTRUCTION

We sampled 100 examples from each dataset and assigned them to three human annotators to label whether they comply with the spatial instructions, taking their average. The results are shown in Table 3.

## I  HUMAN EVALUATION OF DATA QUALITY

We recruited 5 evaluators manually evaluate the quality of our dataset. Each participant was randomly assigned to 100 pairs of data, and they rated them from three dimensions: 1. Editing effect 2. Reasonableness of the instructions 3. Whether the instructions highlight spatial information. The score ranges from 1 to 100. The averaged scores of 5 evaluators are reported. We report the experimental results in Table 4.

| Dataset | Helpfulness | Reasonable | Spatial | Quality |
|---|---|---|---|---|
| IPr2Pr Brooks et al. (2023a) | 55.3 | 87.7 | 23.3 | 57.2 |
| MagicBrush Zhang et al. (2023b) | 68.7 | 89.8 | 45.2 | 62.7 |
| Seed-X Ge et al. (2024) | 66.6 | 88.5 | 37.7 | 68.8 |
| SpatialEdit | 70.4 | 89.3 | 78.3 | 70.2 |

Table 4: Quality assessment of our data engine and other datasets.

## J  DETAILED STATISTICS

For our SpatialEval benchmark, we use 90 samples, 30 from OpenImagesKuznetsova et al. (2020), 30 from COCO2017Lin et al. (2015), 30 from DAQUARMalinowski and Fritz (2014). OpenImages Kuznetsova et al. (2020) is a vast repository of user-uploaded images, offering broad diversity. COCO2017 Lin et al. (2015), on the other hand, specializes in object detection, segmentation, and key point detection tasks. DAQUAR Malinowski and Fritz (2014) features image question-answering tasks, encompassing both direct and inference-based questions.

IPr2Pr Brooks et al. (2023a) dataset consists of 1M CLIP-filtered data pairs, with instructions generated from GPT-3 and images synthesized using Prompt-to-Prompt. MagicBrush Zhang et al. (2023b) annotates 10.5K triples, while MA5k Shi et al. (2021a) contains 24.8K triples aimed at adjusting photo contrast, brightness, or saturation. EVR Tan et al. (2019) gathers 5.7K triples from PhotoshopRequest, while GIER Shi et al. (2020) includes a larger-scale collection of 29.9K triples sourced from online forums.

## K  TEMPLATES

We will open-source the whole code, including the dataset collection code, now we give some sample of templates.

### K.1  SOME TEMPLATES USED IN IMAGE EDITING DATA CONSTRUCTION

```
{
  "direction": [
    "turn the leftmost/rightmost <item> into <style>/<item>",
    "turn the object <relative> <item> into <style>/<item>",
    "turn the <rank> leftmost/rightmost <item> in <items>/<whole>

    into <style>/<item>"
  ],
  "depth": [
    "turn the deepest/farthest <item> into <style>/<item>",
    "turn the nearest <item> into <style>/<item>",
    "turn the <rank> deepest/nearest <item> in <items>/<whole>

    into <style>/<item>"
  ],
  "volumn": [
    "For their real volumn, turn the biggest <item> into <style>

    /<item>/<volumn>",
    "For their real volumn, turn the smallest <item> into

    <style>/<item>/<volumn>",
    "For their real volumn, turn the <rank> biggest/smallest

    <item> in <items>/<whole> into <style>/<item>/<volumn>"
  ],
  "contour": [
```

```
    "For the size of their contour, turn the biggest <item>

    into <style >/<item >/<volumn >",
    "For the size of their contour, turn the smallest <item>

    into <style >/<item >/<volumn >",
    "For the size of their contour, turn the <rank> biggest/smallest <item>

    in <items >/<whole> into <style >/<item >/<volumn >"
  ],
  "movement": [
    "is the objects in the image moving? if so, move it to the position

    they will be at the next moment",
    "move the <direction > <item> to <direction > <move_magnitude >"
  ]
}
```

### K.2    SOME TEMPLATES USED IN VQA DATA CONSTRUCTION

```
{
  "depth": [
    "What is the <rank> tallest?",
    "Is <item> taller/shorter than <item >?",
    "What is the <rank> shortest?"
  ],
  "volumn": [
    "What is the <rank> biggest?",
    "Is <item> bigger/smaller than <item >?",
    "What is the <rank> smallest?"
  ],
  "contour": [
    "What is the <rank> biggest?",
    "Is <item> bigger/smaller than <item >?",
    "What is the <rank> smallest?"
  ],
  "direction":[
    "describe the direction between <item> and <item >",
    "Is <item> <direction > to <item >?",
    "Locate the <item> in image"
  ]
}
```

## L    IMPLEMENTATION DETAILS

We feed the image from the IPr2Pr dataset to the data engine and thus obtain the training set. For the training of MLLM, the learning rate is 1e-6, the pre-trained Llava7B weight is adopted, and the training is conducted on a single A100 GPU with a batch size of 24. For other training tasks, the learning rate is 5e-6, with a warm up of 0.03. Our VQA data consists of 62K question-answer pairs generated for 10K images, while our image editing data contains 25K high-quality data pairs.

For evaluating metrics, Our SpatialEdit is assessed utilizing scores from GPT4V, GPT3.5, and human evaluators. The details of these three evaluation metrics are presented in Appendix N.2.

## M  MORE OPERATIONS OF DATA ENGINE

For the scaling instruction which needs to change the size of an object, the object to be scaled is first removed, then the resized object is placed back in its original position in the image, and stable diffusion is used to blend it naturally. For the visualization instruction which needs to visualize the modified image after some object moves, we use Dragdiffusion Shi et al. (2024) to move the object to a specific position based on the object's trend. For the inpainting instruction which involves replacing one object with another, we employ the inpainting Shi et al. (2024) pipeline of stable diffusion. This process utilizes the object's mask to effectively inpaint the designated area, resulting in object replacement within the image. For the denoting instruction which needs to change the texture or the color of an object at a specific position, we choose to directly change the texture or color of the object within the area enclosed by its mask.

## N  SPATIALEVAL BENCHMARK

### N.1  SOME DATA SAMPLES

In Figure R, we present some samples in various datasets. Considering the need to prevent cheating and data leakage, we do not plan to include the complete test sets in the supplementary materials. However, the SpatialEval benchmark will be made public after the paper is accepted.

### N.2  DETAILED EVALUATION PROCESS

To evaluate the quality of edited images, we propose three evaluation manners. Firstly, we utilize GPT-4V as the judge. The source image and the edited image to be evaluated are fed to GPT-4V. We design a prompt to ask GPT-4V to give a score to the edited image based on its judgment. The prompt is shown in the code we submitted. Secondly, we utilize GPT-3.5 as the judge. Since GPT3.5 is not capable of dealing with image input, so we first use the method in §5.1.1 to extract raw spatial information of the source image and the edited image, and then we obtain two lists of [object, description, trend, position, volume], then two lists are fed the into GPT-3.5, and then we design a prompt to ask GPT-3.5 to judge the results. Thirdly, we ask three human evaluators to evaluate the quality of the edited image. The final score of the edited image is the average of scores of three human evaluators. The range of score is [0,1], with higher scores indicating better quality. The final score is the averaged score of the sample being evaluated.

## O  ABLATION STUDY ON ATTENTION-TUNING LAYERS & VISUALIZATION OF ABLATION STUDY.

We provide visualization results for all our ablation studies (including the investigation of the impact of attention tuning at different attention layers ) in 7.

We found that shallow attention layers might be responsible for localization, as we observed certain changes in the objects to be edited. The middle attention layers seem to handle the background, as models fine-tuned on these layers tend to focus more on the background, while deep attention layers are focused on confirming and applying the edits. We believe these three components should work collaboratively, so our attention tuning is not restricted to specific depths.

## P  CLARIFY THE MOTIVATION & CONTRIBUTION

If we rewrite the conditional generation probability of the spatial editing problem in the form of a Bayesian formula, we have:

$$
\begin{aligned}
\nabla \log p\left(\boldsymbol{x} \mid y\right) &= \nabla \log \left(\frac{p\left(\boldsymbol{x}\right) p\left(y \mid \boldsymbol{x}\right)}{p(y)}\right) \\
&= \nabla \log p\left(\boldsymbol{x}\right) + \nabla \log p\left(y \mid \boldsymbol{x}\right) - \nabla \log p(y) \\
&= \nabla \log p\left(\boldsymbol{x}\right) + \nabla \log p\left(y \mid \boldsymbol{x}\right)
\end{aligned}
\tag{31}
$$

|  | L1 $\downarrow$ | CLIP-I $\uparrow$ | LPIPS $\downarrow$ |
|---|---|---|---|
| MGIE | 0.126 | 0.789 | 0.347 |
| SmartEdit | 0.176 | 0.683 | 0.446 |
| InstructDiffusion | 0.144 | 0.798 | 0.305 |
| SpatialEdit-LoRA | 0.202 | 0.649 | 0.474 |
| SpatialEdit | 0.072 | 0.860 | 0.192 |

Table 5: Evaluation results on SpatialBench.

The conditional generation results are influenced by an implicit classifier, which can be represented by the MLLM part.

This highlights the impact of the implicit classifier on the final editing results. Specifically, the model needs to "understand" the spatial relationships within the image to generate effectively, which explains the motivation behind training MLLM in our first stage.

Note that in our derivation of diffusion process, the gradient of the posterior of the $y$ is 0. In fact, in the optimal state achievable by the model (regardless of how sophisticated the model architecture is), the constraint of $y$ always exists, and we have:

$$\log p\left(\boldsymbol{x} \mid y\right)^{*} = \log p\left(\boldsymbol{x}\right)^{*} + \log p\left(y \mid \boldsymbol{x}\right)^{*} - \log p(y) \tag{32}$$

Moreover, the experiments in our paper demonstrate that the current dataset lacks sufficient spatial instructions. Therefore, it is crucial to develop a data engine to generate spatial instructions.

We believe that each module in our method is well-motivated. The primary technical contributions of our approach lie in the two-stage training strategy and the data engine. Given the significant deficiencies in the current dataset, we argue that architectural improvements alone cannot fundamentally address this issue. We kindly request the reviewers to reassess the novelty of our work.

In summary, the derivation of Theorem 1 highlights the requirements for the amount of data, while the above derivation emphasizes the requirements for the content of the data.

## Q  METRIC ABOUT SPATIALBENCH

Our human annotators used computer tools to provide ground truth for all edits, and we offer three metrics: L1, CLIP-I, and LPIPS. The results show that our SpatialEdit still achieves SOTA performance on the SpatialBench.

## R  COMPARE WITH MORE MODELS

Although InstructDiffusion Geng et al. (2024) and SmartEdit Huang et al. (2024) were trained on significantly more data than ours, we still compared our method with their open-source checkpoints.

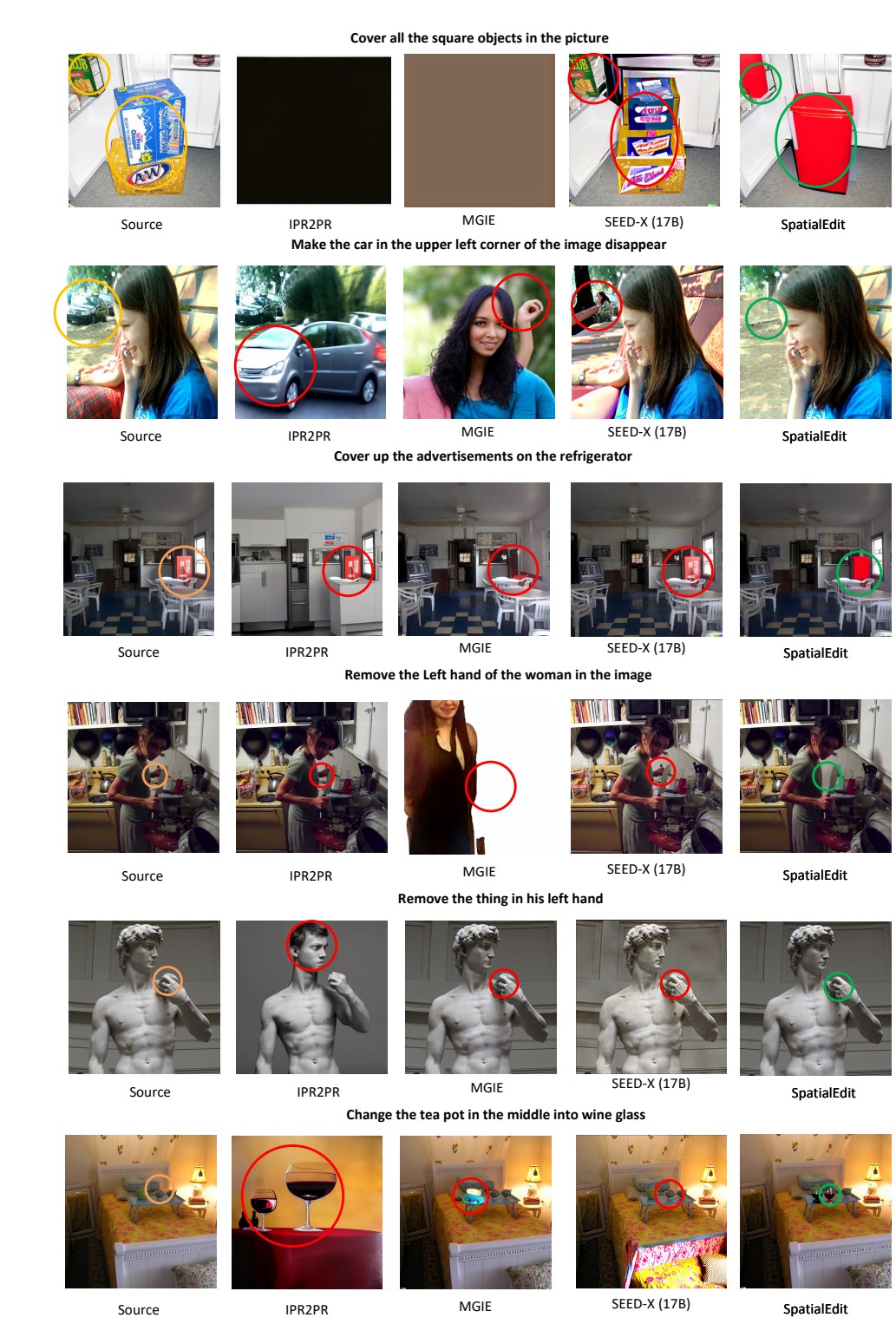

Figure 4: More Examples of Image Editing Results. The orange circles represent areas that need editing, the red circles indicate incorrect edits, and the green circles represent correct edits. We can see that with complex prompts, especially spatial prompts, our model is capable of correctly editing images and demonstrates a profound understanding of spatial information, something that even a 17B model finds challenging.

**COCO2017**

"Turn the clothes color of closest people into blue"

"Make the bread under the beef"

"Turn the leftmost pillow into green"

**DAQUAR**

"Change the object between two chairs into tree"

"Make the TV smaller"

"Make the isolated chair smaller"

**OpenImage**

Draw a bird on the clothes of the third leftmost people

"Draw the next time they will be"

"Make the cow in middle smaller"

Figure 5: Test samples (including the image from different datasets and the instruction) in SpatialEval benchmarks.

| Origin | MGIE | SmartEdit | InstructDiffusion | SpatialEdit |

**Instruction: Make the water glowing.**

**Instruction: Remove the chair closest to you that you see in front of you.**

**Instruction: Have the plane on the right perform a barrel roll maneuver.**

Figure 6: New Comparisons with more baselines.

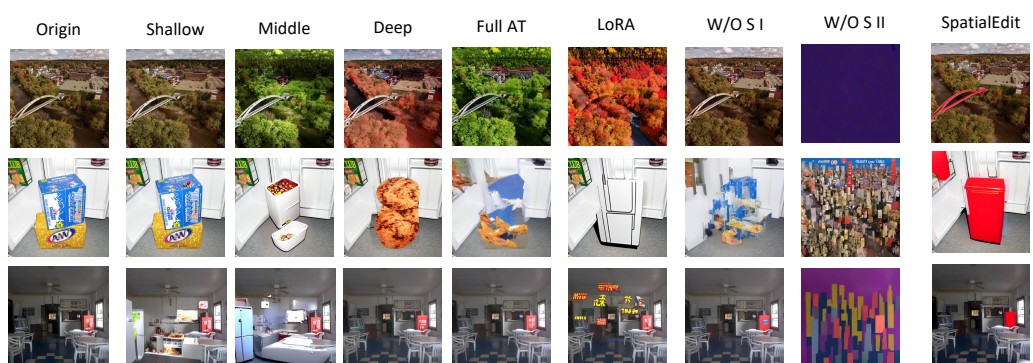

Figure 7: Visualization of ablation study.

