# OpenReview forum: "SpatialEdit: Unlocking the  Spatial Capability in Multimodal Large Language Model Driven Image Editing"
_ICLR.cc/2025/Conference — Submitted to ICLR 2025_

### Official Review · Reviewer_Y5q6 · 2024-10-16

**Soundness:** 2
**Presentation:** 3
**Contribution:** 2
**Rating:** 5
**Confidence:** 4

**Summary:**

This paper highlights the issue of understanding complex instruction problems in the MLLM editing task. The theoretical analysis presented in the paper suggests that both the training strategy and the model aggregation method within the current paradigm may contribute to these unsatisfactory results. Consequently, the authors propose a data engine capable of automatically generating spatial VQA (Visual Question Answering) data and spatial image editing data from a given image to prevent potential overfitting. Additionally, two datasets are utilized during the training stage for MLLM attention tuning, thereby enhancing the capabilities of spatial editing. To assess the spatial editing ability, the paper establishes a benchmark and achieves state-of-the-art (SOTA) results on several key metrics.

**Strengths:**

-  The theoretical analysis presented in this paper is intriguing, employing Rademacher complexity to demonstrate the potential risk of overfitting and utilizing convex divergence to highlight the necessity of spatial conditions for generalization.
- The data engine's ability to automatically generate spatial VQA (Visual Question Answering) data and spatial image editing data from a given image alleviates the burden of manual data preparation.
- The two-stage training approach contributes to the paper achieving state-of-the-art (SOTA) performance on the newly created benchmark. Specifically, the attention-tuning is more important for spatial understanding.

**Weaknesses:**

- The visual evaluation for the ablation experiments is lacking, which makes it difficult to ascertain the significance of attention-tuning and stage II training. Including such evaluations would provide clearer insights into the contributions of these components.

- It is necessary to clarify the differences between the two-stage training method and other approaches. If the explicit spatial condition is crucial, it should be investigated whether other methods could also benefit from training on a spatial editing dataset to achieve better results.

- SmartEdit[1] has previously proposed complex instruction editing with MLLM and demonstrated spatial editing capabilities. The experiments in this paper would benefit from a comparison with SmartEdit to determine if the current approach offers any advantages or unique insights.




[1] Huang, Yuzhou, et al. "Smartedit: Exploring complex instruction-based image editing with multimodal large language models." Proceedings of the IEEE/CVF Conference on Computer Vision and Pattern Recognition. 2024.

**Questions:**

- The paper makes certain assumptions that require validation. For instance, is it accurate to consider MLLM as Lipschitz continuous?

- In Figures I and II, the choice of 'red' for editing seems discordant with the overall color scheme. Could this be due to color saturation, contrast issues, or another graphical aspect? It would be beneficial to address this discrepancy to ensure the visual presentation is cohesive and effective. Can the authors suggest solutions or adjustments to improve the visual harmony?

- The paper would benefit from including more visual comparison results.

---

### Official Review · Reviewer_pDiM · 2024-11-02

**Soundness:** 3
**Presentation:** 3
**Contribution:** 2
**Rating:** 5
**Confidence:** 4

**Summary:**

The paper addresses the challenge of instructional image editing, particularly with respect to spatial-aware instructions. It has been observed that existing MLLM-based methods for image editing perform unsatisfactory when it comes to understanding complex instructions. The paper theoretically analyzes these limitations, attributing them to potential overfitting and ineffective training strategies. In response, the authors propose a data engine and a two-stage training strategy to mitigate these issues. To accurately assess model performance, the paper also introduces a benchmark specifically designed for spatial image editing tasks.

**Strengths:**

1. The paper is well-structured and easy to follow.
2. The issue of instructional image editing is highly relevant to real-world applications, and the observation of the unsatisfactory performance of existing models on spatial instructions is valuable for further research.
3. The paper effectively analyzes the limitations of current models and introduces a data engine and training approach to address these issues, which is fundamentally sound.
4. The proposed method achieves state-of-the-art performance on the newly introduced benchmark.

**Weaknesses:**

1. The newly proposed dataset for benchmarking models is too small-scale, comprising only 90 images, to adequately assess model performance. Additionally, the dataset lacks ground-truth images, and model candidates can only be evaluated by GPT or human annotators, which may introduce subjective bias. Consequently, the evaluation results may not be sufficiently convincing, potentially diminishing its value to the community. Furthermore, more qualitative comparisons should be included. Additionally, why does the method differ between Table 1 and Table 2?
2. The introduced attention tuning is intriguing. However, since it appears to be the sole technical contribution of this work, more in-depth ablation studies are necessary. For instance, how does tuning different attention layers affect performance, and how does it compare to the widely adopted LoRA tuning?
3. The paper lacks necessary discussions and comparisons with existing image editing works, such as SmartEdit [A].
> [A] SmartEdit: Exploring Complex Instruction-based Image Editing with Multimodal Large Language Models. CVPR 2024.

**Questions:**

Please refer to the Weaknesses section.

---

> ### Comment · Reviewer_pDiM · 2024-11-27
>
> I'd like to thank authors for the response. However, I have decided to maintain the initial rating as some concerns raised were not addressed and remain unresolved. For instance, the newly proposed benchmark is too small to properly evaluate the models. There is a discrepancy in the methods compared in Table 1 and Table 2, which may lead to concerns about the proposed model's robustness across different benchmarks. Furthermore, the authors have only provided visualizations for ablation studies regarding different tuning strategies, which is insufficient. Qualitative results should also be incorporated.

---

### Official Review · Reviewer_aHJt · 2024-11-03

**Soundness:** 3
**Presentation:** 2
**Contribution:** 2
**Rating:** 5
**Confidence:** 5

**Summary:**

This paper introduces SpatialEdit, an approach to address the limitations of current MLLM-based methods in handling spatial editing instructions. The authors propose a data engine to generate high-quality spatial editing data, a two-stage training approach with attention tuning, and a new benchmark called SpatialEval to evaluate spatial editing performance.

**Strengths:**

1. The paper suggests using the VQA dataset to enhance spatial understanding capabilities.
2. It identifies the limitations of MLLM-based methods through both theoretical and experimental analysis. The proposed data engine and two-stage training approach with attention tuning are effective solutions.
3. The paper is generally well-written and logically structured.
4. Improving spatial editing capabilities in image editing systems is an important problem with practical applications. The proposed method shows improvements over existing approaches, potentially advancing the field of instruction-guided image editing.

**Weaknesses:**

1. How does the attention tuning method compare to other fine-tuning approaches for MLLMs, such as full fine-tuning and LoRA fine-tuning?
2. The theoretical analysis for the first limitation—only tuning the adapter leads to overfitting issues—seems not very insightful. The analysis is broad and the insight appears well-known. Researchers typically use LoRA tuning to address such problems.
3. Tuning more parameters, including attention, tends to increase the risk of overfitting with limited data. Why not use just LoRA tuning as is commonly practiced?
4. EMU-Edit [1] does not use the MLLM model but relies on a classic text encoder, yet it still demonstrates spatial understanding and editing capabilities. This raises the question of whether incorporating MLLM in instruction-based editing is necessary. What are the authors' thoughts on this?
5. The paper lacks a discussion of related work and does not compare similarities and differences with other methods such as EMU-Edit [1], SmartEdit [2], and InstructDiffusion [3].
6. Evaluation: It is suggested to use segmentation datasets such as the ADE20K evaluation dataset to assess edit performance, providing a better measure of generalizability and enabling fair comparisons.
7. The results in Supp. Fig4 show black and brown colors, which is unusual and requires explanation.
8. The paper lacks a discussion of limitations and failure cases.
9. Better evaluation is needed. The paper states that "three human evaluators assess the quality of the edited image," but the focus should be on edit accuracy rather than just the quality of the edited image.
10. The paper mentions that SpatialEdit uses Llava-7B as the backbone MLLM. Have the authors experimented with other MLLM architectures or sizes? How sensitive is the method to the choice of the backbone model?

[1] EMU-Edit: Sheynin, Shelly, et al. "Emu edit: Precise image editing via recognition and generation tasks." *Proceedings of the IEEE/CVF Conference on Computer Vision and Pattern Recognition*. 2024.
[2] SmartEdit: Huang, Yuzhou, et al. "Smartedit: Exploring complex instruction-based image editing with multimodal large language models." *Proceedings of the IEEE/CVF Conference on Computer Vision and Pattern Recognition*. 2024.

[3]InstructDiffusion: Geng, Zigang, et al. "Instructdiffusion: A generalist modeling interface for vision tasks." *Proceedings of the IEEE/CVF Conference on Computer Vision and Pattern Recognition*. 2024.

**Questions:**

See the [Weaknesses] part

---

### Official Review · Reviewer_jLTr · 2024-11-04

**Soundness:** 2
**Presentation:** 2
**Contribution:** 1
**Rating:** 3
**Confidence:** 4

**Summary:**

This paper aims to improve the current Image Editing methods, in particular focusing on spatial understanding. To that end, the paper analyses and reveals the image editing methods which are based on Large Multimodal Models (LMMs), often failing because of the LMMs' spatial understanding abilities.
The paper also introduces a novel framework that achieved the best results on the proposed benchmark.

**Strengths:**

- Overall, I think studies to understand more specific abilities of Large Multimodal Models are useful, especially as these models are now used for many downstream tasks (e.g., Image Editing).
- The paper has a generally clear structure.

**Weaknesses:**

- There are many typos in the paper, and the writing is somewhat difficult to read (e.g., Line 42-44, Line 123, etc.).
- While I agree that understanding the limitations of Large Multimodal Models is a good direction, I feel the connection between Image Editing and Large Multimodal Models is weak. I’d suggest the authors consolidate the paper by focusing on either task (e.g., if "Understanding and Improving the Spatial Understanding of Large Multimodal Models" is the focus, then Image Editing could be one application). In the current version, it is not clear whether the paper’s focus is on Image Editing or the Spatial Abilities of LMMs.
- The experiments section is not thorough. As far as I understand, the proposed framework will fine-tune the LMMs. I’d suggest including “forgetting” in the evaluation.

**Questions:**

I’d be happy to adjust my score if the authors can provide a convincing argument or a detailed explanation.

---

### Meta-Review · Area_Chair_gZmT · 2024-12-19

**Metareview:**

This paper proposes a spatial-aware image editing framework SpatialEdit, addressing the limitations of previous image editing approaches in spatial understanding. Four reviewers consistently give negative scores with major concerns on limited contribution and insufficient experimental validation. The rebuttal did not convince the reviewers on the proposed issues and questions. Therefore, the area chair agrees with the reviewers and would recommend rejecting the paper.

**Additional Comments On Reviewer Discussion:**

Reviewers proposed various questions. While some of them are addressed during rebuttal, there remains significant number of questions that the authors did not give convincing answers. 3 out of 4 reviewers read the rebuttal and responded, but none of the reviewers increased the scores during the rebuttal and discussion period.

---

### Decision · Program_Chairs · 2025-01-22

Reject